# Gene Therapy for Acquired and Genetic Cholestasis

**DOI:** 10.3390/biomedicines10061238

**Published:** 2022-05-26

**Authors:** Javier Martínez-García, Angie Molina, Gloria González-Aseguinolaza, Nicholas D. Weber, Cristian Smerdou

**Affiliations:** 1Division of Gene Therapy and Regulation of Gene Expression, Cima Universidad de Navarra, 31008 Pamplona, Spain; jmartinez.71@alumni.unav.es (J.M.-G.); amolinad@unav.es (A.M.); ggonzalez@vivet-therapeutics.com (G.G.-A.); 2Instituto de Investigación Sanitaria de Navarra (IdISNA), 31008 Pamplona, Spain; 3Vivet Therapeutics S.L., 31008 Pamplona, Spain

**Keywords:** cholestatic diseases, gene therapy, AAV, PFIC

## Abstract

Cholestatic diseases can be caused by the dysfunction of transporters involved in hepatobiliary circulation. Although pharmacological treatments constitute the current standard of care for these diseases, none are curative, with liver transplantation being the only long-term solution for severe cholestasis, albeit with many disadvantages. Liver-directed gene therapy has shown promising results in clinical trials for genetic diseases, and it could constitute a potential new therapeutic approach for cholestatic diseases. Many preclinical gene therapy studies have shown positive results in animal models of both acquired and genetic cholestasis. The delivery of genes that reduce apoptosis or fibrosis or improve bile flow has shown therapeutic effects in rodents in which cholestasis was induced by drugs or bile duct ligation. Most studies targeting inherited cholestasis, such as progressive familial intrahepatic cholestasis (PFIC), have focused on supplementing a correct version of a mutated gene to the liver using viral or non-viral vectors in order to achieve expression of the therapeutic protein. These strategies have generated promising results in treating PFIC3 in mouse models of the disease. However, important challenges remain in translating this therapy to the clinic, as well as in developing gene therapy strategies for other types of acquired and genetic cholestasis.

## 1. Cholestatic Diseases

Cholestatic diseases are based on bile dysfunction due to defects affecting bile synthesis or secretion. These processes involve a wide range of enzymes and membrane transporters involved in hepatobiliary circulation. According to its origin, cholestasis can be classified into two main groups: acquired cholestasis and genetic cholestasis [1].

### 1.1. Acquired Cholestasis

Most cholestatic diseases are acquired, presenting a dysregulation of the hepatobiliary transporters as a consequence of an adaptive and protective response to bile acid (BA) accumulation in the liver. This regulation is multifactorial, involving different elements such as hormones, BAs, proinflammatory cytokines, and drugs. These different factors mediate the activation of transcription factors that regulate the expression of export pumps, which promote the reduction of intracellular BAs by their excretion in the urine, resulting in the detoxification of the liver [2]. Acquired cholestatic diseases include primary biliary cholangitis (PBC), primary sclerosing cholangitis (PSC), intrahepatic cholestasis of pregnancy (ICP), biliary atresia, drug-induced cholestasis, and inflammation-mediated cholestasis [1,3]. 

PBC and PSC are classified as autoimmune diseases of the hepatobiliary system, characterized by the presence of antimitochondrial antibodies, portal inflammation, and an immune-mediated destruction of intra- and extra-hepatic bile ducts [4,5]. Clinical manifestations vary widely, from asymptomatic to end-stage biliary cirrhosis. The pathogenesis of the disease is multifactorial, involving genetic, epigenetic, and environmental factors [4,6]. 

ICP, which is the most common disorder of the hepatobiliary system, is characterized by high serum BA levels in the third trimester of pregnancy that cause severe pruritus. In the development of this cholestatic disorder, high levels of gestational hormones, such as estrogen and progesterone, play a major causative role, while genetic factors may also be involved. Although symptoms disappear after childbirth, the biliary disorder can often recur during future pregnancies [7].

Biliary atresia is a rare liver disease affecting the bile ducts, resulting in the main cause of neonatal cholestasis. The etiology of this biliary disorder is unknown. In some cases, the origin is thought to be due to an exacerbated autoimmune response in the bile duct epithelium as a consequence of a viral infection or due to toxin-induced injury after birth [8]. In other cases, it is thought to be due to a malformation of the bile ducts during gestation. However, it is known that an early diagnosis allows for better outcomes after surgery [9].

Finally, drug- and inflammation-induced cholestasis are closely related. Both drugs and proinflammatory agents can induce cholestasis following inhibition of hepatobiliary transporters but rarely result in severe liver injury. These types of cholestasis have an immunological origin mediated by proinflammatory cytokines directed against the bile duct epithelium that can alter BA secretion [10]. 

### 1.2. Inherited Cholestasis

Genetic cholestasis, which represents a minority of all cholestatic disorders, includes different types of progressive familial intrahepatic cholestasis (PFIC) associated with mutations in relevant channel transporters of the hepatobiliary system. PFIC is a heterogeneous group of autosomal recessively inherited monogenic disorders with a low incidence of 1:50,000–100,000 births worldwide, representing approximately 15% of all cases of neonatal cholestasis [11]. These cholestatic syndromes are characterized by an early onset of the disease, usually in infancy, associated with clinical manifestations such as pruritus, jaundice, malabsorption of fat and fat-soluble vitamins, and hepatomegaly [11]. PFIC is associated with several liver complications, such as portal hypertension and cirrhosis, and can progress to end-stage liver disease and liver failure between childhood and adulthood. Depending on the type of PFIC, extrahepatic clinical manifestations or hepatocellular carcinoma (HCC) may occur [12]. The most common biochemical features of this group of hepatobiliary diseases are increased serum BAs and bilirubin [11]. Depending on their genetic origin, PFICs can be classified into six types. Mutations in *ATP8B1*, *ABCB11*, *ABCB4*, tight junction protein 2 (*TJP2*), *NR1H4*, and Myosin VB (*MYO5B*) genes are known to be the cause of PFIC 1-6 types, respectively (Figure 1). In PFIC1, mutations in the familial intrahepatic cholestasis 1 (*FIC1*) gene cause the loss of the asymmetric distribution of phospholipid content in the canalicular membrane, leading to membrane destabilization and reduced BA transport, resulting in their accumulation in hepatocytes, causing cholestasis. Mutations in the *ABCB11* gene can result in PFIC2 due to the absence of a functional bile salt export pump (BSEP) protein, which also leads to toxic accumulation of BA in hepatocytes. In PFIC3, mutations in *ABCB4* cause multidrug resistance protein 3 (MDR3, ABCB4) deficiency, which results in low levels of phosphatidylcholine (PC) in the bile, which is needed to form micelles and neutralize the toxicity of hydrophobic BAs, resulting in damage to the epithelium of bile canaliculi. Mutations in *TJP2* lead to the misdistribution of claudin tight junction in canaliculi, resulting in bile leakage and subsequently in PFIC4. PFIC5 is due to mutations in the *NR1H4* gene that cause deficiency in farnesoid X receptor (FXR), resulting in a reduction of BSEP and ABCB4 expression and the accumulation of toxic BAs in the hepatocytes. Finally, mutations in *MYO5B* interfere with the processing of normal intracellular trafficking of BSEP, reducing its expression and activity at the canalicular membrane, which results in the accumulation of toxic BAs in hepatocytes, giving rise to PFIC6 [13]. Different disease characteristics such as the age of onset, severity, and the manifestation of specific complications and serum markers vary between PFIC types [12,13]. 

The role of BSEP in the functioning of the hepatobiliary system is very important, as mutations in different genes involved in BA metabolism and transport, such as *ABCB11*, *NR1H4*, and *MYO5B* causing its deficiency, cause PFIC [14,15,16]. In addition, depending on the severity of the disease, inherited intrahepatic cholestasis resulting from mutations in *ATP8B1* or *ABCB11* can be classified as either PFIC1 or 2, respectively, or benign recurrent intrahepatic cholestasis (BRIC) 1 or 2, respectively. Sometimes it is clinically difficult to discern between PFIC and BRIC because, in both cases, patients may present mild cholestasis with long-term complications [17]. In addition, some missense mutations in less conserved regions of the *ABCB11* and *ABCB4* genes promote the development of more moderate variants of cholestasis such as BRIC2, ICP, cholesterol cholelithiasis, drug-induced cholestasis, adult biliary cirrhosis, transient neonatal cholestasis, and others [18,19]. In addition, mutations in cholangiocyte transporter genes (e.g., the cystic fibrosis transmembrane conductance regulator (*CFTR*) gene) can cause cholestasis. In fact, a direct association between cystic fibrosis and cholestatic conditions, such as bile duct complications, gallstones, and primary sclerosing cholangitis, has been observed due to mutations in *CFTR* [20]. Other genetic multisystemic diseases associated with cholestatic disorders include Alagille syndrome (ALGS) and cerebrotendinous xanthomatosis (CTX). ALGS arises due to mutations in genes involved in the Notch signaling pathway, such as *JAG1* and *NOTCH2*, and the majority of patients present cholestasis and a deficiency of bile ducts [21]. CTX is caused by mutations in the *CYP27A1* gene, resulting in impaired BA biosynthesis and the accumulation of toxic metabolites. Although liver damage is not common in all CTX patients, some cases of severe infantile cholestasis have been reported [22].

## 2. Current Treatments

### 2.1. Surgical Procedures: Hurdles and Limitations

Currently, therapeutic approaches for cholestatic disorders are limited, with liver transplantation being the only curative strategy for the more severe syndromes [23,24]. However, liver transplantation has numerous limitations, such as organ failure, donor shortage, limited organ viability, the requirement of life-long immunosuppression, and immunological rejection [25]. For inherited diseases, such as some types of PFIC, liver transplantation is considered for end-stage patients with severe complications, such as hepatocellular carcinoma (HCC), hepatic steatosis, and liver cirrhosis. Orthotopic transplantation successfully improves cholestasis and related symptoms in 3–5 years [12,26]. However, liver transplant has been shown to be associated with the development of circulating anti-BSEP antibodies in a small fraction of transplanted PFIC2 patients, resulting in the rejection of the transplanted organ [27,28]. Moreover, this approach is only partially effective for cholestatic diseases with extrahepatic manifestations, such as PFIC1.

A therapeutic alternative prior to liver transplantation is surgical treatment aiming to interrupt the enterohepatic circulation, including procedures, such as partial internal biliary diversion (PIBD), ileal exclusion, and partial external biliary diversion (PEBD), that lead to lower BA levels, less pruritus, and even reversal of hepatic fibrosis [29,30]. However, complications such as stoma bag-associated difficulties (e.g., dehydration or leakage) have been reported [30]. For treatment of hereditary cholestatic diseases, biliary diversion has been found to be more effective in PFIC2 patients with residual BSEP activity, while for PFIC3 patients it is usually done late in the disease process, making it hard to prevent disease progression [31,32]. Therefore, there is an urgent need to seek alternative therapeutic approaches to liver transplants and surgical approaches. However, there is room for hope since the increased understanding of the mechanisms leading to genetic and acquired cholestatic diseases has opened the window to develop new drug and gene therapies for the treatment of these disorders.

### 2.2. Pharmacological Therapies

Drug therapies are considered first-line treatments for cholestatic diseases. The main strategies in the pipeline are based on FXR agonists and inhibitors of BA uptake transporters in the enterohepatic circulation [33,34]. 

#### 2.2.1. FXR Agonists

In recent years, the use of selective FXR agonists, such as ursodeoxycholic acid (UDCA), has been the first option to treat cholestatic disorders. UDCA, a hydrophilic BA, reduces the hydrophobic pool of toxic BAs in hepatocytes as well as the detergent properties of bile in the bile canaliculi (Figure 2A). Currently, beneficial effects of UDCA have been reported in patients with ICP, PBC, and PFIC3, especially at the early stages of these diseases [35,36], although approximately 50% of the PFIC3 and PBC patients did not respond or had an incomplete response [19,37]. It has also been observed that PFIC3 patients with milder forms of ABCB4 deficiency respond better to UDCA treatment [38]. In contrast, this treatment fails to offer any symptomatic improvement for the majority of patients with PFIC2 or PSC [39,40]. On the other hand, UDCA-derived BAs such as 24-norursodeoxycholic acid (Nor-UDCA) or its taurine conjugate (TUDCA) have also shown potential as therapeutic agents for these liver diseases [35]. Nor-UDCA has shown improvement in serum disease biomarkers such as transaminases and alkaline phosphatase (ALP) levels in patients with PSC [41], although larger studies are needed to establish its real efficacy [42]. Currently, there is one clinical trial evaluating its use in PSC patients (NCT01755507). A recent study has shown that TUDCA was able to normalize serum ALP values in PBC patients [43]. Another FXR agonist with therapeutic potential in the treatment of cholestatic diseases is the semi-synthetic BA, obeticholic acid (OCA). Two phase II studies in PBC and PSC patients demonstrated the safety and beneficial effect of OCA in reducing serum ALP levels [44,45] and, in fact, OCA has been approved as an alternative treatment for patients with PBC who do not respond to UDCA [46]. In addition, a recent study showed that OCA was able to reduce liver damage in a mouse model of PFIC2 [47]. Despite these promising results, its use in cholestatic patients has been associated with severe pruritus, which would make it difficult to be approved as a therapy for PFIC, in which pruritus is one of the main symptoms of concern [48]. Similarly, the non-steroidal FXR agonist cilofexor, which has been reported to lead to significant improvements in cholestasis markers in PSC patients [49], may cause pruritus in a dose-dependent manner as a side effect and is not recommended for certain cholestatic disorders [50].

Altogether, these data indicate that the identification and development of new and more efficient FXR agonists represents a very interesting area of investigation for the improved clinical management of cholestatic diseases (Table 1) [51,52].

#### 2.2.2. Inhibitors of Bile Acid Uptake Transporters

Recently, there has been great interest in developing drugs that are able to interrupt the enterohepatic circulation in a non-invasive manner for cholestatic disorders. The four transporters that allow circulation of BAs between the liver and intestine are the apical bile salt transporter (ASBT, also known as IBAT for ileal bile acid transporter), BSEP, the sodium-taurocholate cotransporter polypeptide (NTCP) and the basolateral organic solute transporter (OST) [1]. The inhibition of BSEP and OST transporters is not an option as this would result in toxic accumulation of BAs in hepatocytes and enterocytes, respectively [53,54]. In contrast, pharmacological inhibition of the hepatic transporter NTCP results in a well-tolerated increase of BAs in plasma and a subsequent decrease in the liver (Figure 2B) [55]. In fact, recent studies have shown the hepatoprotective effect of NTCP inhibition, resulting in attenuation of cholestasis [56]. ASBT inhibitors prevent the reabsorption of BAs in enterocytes and their recirculation to the liver, favoring their excretion in feces (Figure 2C). ASBT antagonists currently being tested in clinical trials include odevixibat (A4250, Albireo, Boston, MA, USA), maralixibat (LUM001, Mirum Pharmaceuticals, Foster City, CA, USA), elobixibat (A3309, Albireo), linerixibat (GSK2330672, Glaxo Smith Kline, Brentford, United Kingdom) and volixibat (SHP626, Mirum Pharmaceuticals) (Table 1) [57,58]. Several preclinical studies and clinical trials have shown high safety profiles for all these compounds with limited adverse effects outside the gastrointestinal tract and a high specificity for ASBT when orally administered. The observed therapeutic effects include a decrease of BAs in the liver and serum, reduction in pruritus, liver inflammation, and liver fibrosis [57,58]. In 2021, odevixibat was approved for clinical use in PFIC patients by the US Food and Drug Administration (FDA) and European Medicines Agency (EMA). Moreover, its safety and efficacy for treatment of other cholestatic diseases, such as ALGS, are being evaluated [59]. Maralixibat has also been evaluated in PBC and PSC, but clinical trials were discontinued because this treatment did not improve pruritus compared to placebo [60]. Recently, maralixibat was approved for clinical use for ALGS patients by the FDA [61]. However, its use for other cholestatic diseases, such as PFIC1-4, is currently under evaluation by the EMA [62]. 

#### 2.2.3. Other Pharmacotherapeutic Agents 

Further additional pharmacotherapeutic approaches for the treatment of cholestatic disorders are being explored. Peroxisome proliferator-activated receptor (PPAR) agonists and fibroblast growth factor (FGF) analogues have been shown to be effective for diseases such as PBC and PSC [63]. Activators of FXR transcriptional regulators, such as sirtuin 1, have been shown to alleviate cholestatic liver injury in mice with BA-induced cholestasis by increasing the hydrophilic character of the hepatic BA composition and decreasing plasma BA concentration [64]. The use of antifibrogenic and anti-inflammatory therapeutic agents, such as inhibitors of histone deacetylases and phosphodiesterase 5, led to reduced fibrosis and liver damage in a PFIC3 mouse model [65]. Finally, ABC transporter enhancers, such as ivacaftor, may rescue the functionality of canalicular membrane transporters implicated in cholestatic disorders, including BSEP. Thus, PFIC2 patients may benefit from this type of pharmacological treatment [66]. The use of fibrates, such as the PPAR agonists bezafibrate, fenofibrate, and elafibranor (Table 1), could also be beneficial for the treatment of PBC patients who do not respond to UDCA [67].

Although the pharmacological strategies mentioned above significantly improved the pathology of cholestatic diseases and the quality of life of the patients [63], they do not represent a definitive cure for hepatobiliary dysfunction. For this reason, the development of new strategies, such as cell and gene therapy, that allow stable, long-term correction of these diseases is highly desired. In the following section, we will focus on gene therapy strategies tested in preclinical models of cholestatic diseases.

## 3. Gene Therapy

Gene therapy involves the addition, removal, or modification of the genetic material of an individual in order to treat a disease [83]. Its efficacy depends on successful delivery to target cells, for which vectors (viral and non-viral) are utilized. Viral vectors are based on modified viruses, such as adenoviruses (Adv), adeno-associated viruses (AAV), retroviruses, and lentiviruses, among others, which have proven to be very effective for gene delivery, although they present some drawbacks such as immunogenicity and limitations in cargo size. Non-viral vectors, such as polymeric or lipid nanoparticles (LNP), unlike viral vectors, do not achieve delivery to the cell nucleus and induce much more transient transgene expression, but have a better safety profile, are not limited by packaging restrictions, and offer several advantages in manufacturability and shelf-life. Recently, non-viral vectors have shown a high degree of efficacy as demonstrated by the COVID-19 vaccines based on messenger RNA (mRNA)-containing LNPs [84]. 

Gene therapy has emerged as a promising approach to achieve safe, stable, and efficient long-term correction for a wide range of genetic diseases [85], including monogenic liver disorders, for which liver transplantation remains the only cure [86], as well as acquired liver diseases [87]. Viral and non-viral vectors have shown promising therapeutic results in numerous clinically relevant animal models, as well as in a large number of clinical trials [88,89]. The fact that more than a dozen gene therapy products have been approved by the FDA and EMA, albeit only three for liver gene therapy, is a promising sign for the future application of this technology for liver disorders [90,91].

### 3.1. Gene Therapy for Acquired Cholestasis

Since no definitive treatment has yet been developed for some acquired hepatic cholestasis, such as PBC and PSC, there is a great need to identify novel therapeutic alternatives that can reduce fibrogenesis and potentially prevent the development of chronic liver injury, making genetic-based treatments an attractive strategy to achieve sustained long-term therapeutic effects.

To generate animal models of acquired cholestatic disorders, interventions including bile duct ligation (BDL) and the induction of cholestasis by drugs, such as estrogens and carbon tetrachloride (CCL_4_), have been utilized [92]. The development of cholestasis involves several processes including: cellular apoptosis, production of proinflammatory cytokines, and fibrogenesis that ultimately leads to biliary impairment [93].

Gene therapy approaches for acquired cholestasis have been addressed to mitigate liver damage by reducing apoptosis and fibrosis and improving bile formation (Figure 3). Next, we will describe the most relevant gene therapy strategies described so far. 

#### 3.1.1. Apoptosis Attenuation

One of the main targets for gene therapy of acquired liver disorders is the reduction of hepatocyte apoptosis. Hydrodynamic-based gene delivery to the liver of an insulin-like growth factor 1 (IGF-1)-expressing plasmid has demonstrated attenuation of hepatocellular apoptosis and liver injury in rats with BDL. IGF-1 promotes amelioration of cholestatic disease through activation of the phosphatidylinositol-3-kinase pathway, the inhibition of glycogen synthase kinase-3 beta, and the blockade of caspase-9 cleavage. Additionally, inactivation of hepatic stellate cells has been observed, which may explain the notable improvement in the degree of liver fibrosis [94]. 

#### 3.1.2. Reduction of Mitochondrial Oxidative Stress 

Reducing oxidative stress has been shown to be a therapeutic target for acquired liver cholestasis. For example, Adv-mediated mitochondrial superoxide dismutase (SOD) gene delivery leads to a reduction in liver injury by avoiding the formation of oxygen free radicals derived from the accumulation of hydrophobic BAs and preventing the release of proinflammatory cytokines, such as TNFα and TGF-β, in mice with BDL [95]. Similarly, administration of Adv vectors expressing an inhibitor gene of proinflammatory cytokine signaling like collagen triple helix repeat containing-1 (Cthrc-1) has shown a reduction of liver fibrosis in mice subjected to BDL and drug-mediated cholestasis through the inhibition of TGF-β signaling caused by the accelerating degradation of phospho-Smad3 [96].

#### 3.1.3. Anti-Fibrotic Therapies 

Anti-fibrotic therapies for cholestatic disorders via reducing pro-inflammatory factors tend to promote collagen degradation and thus reduce the degree of liver fibrosis. Adv vectors expressing the urokinase-plasminogen activator (uPA) gene resulted in a slight reduction of liver fibrosis, leading to a partial improvement of liver histology in rats with BDL associated with the activation of metalloproteinases that trigger collagen degradation [97,98]. Additionally, AAV vectors that allow hepatic expression of angiotensin-converting enzyme (ACE2) provided a sustained anti-fibrotic effect in different animal models of BDL and drug-induced cholestasis [99]. A different strategy to fight fibrosis is based on the gene delivery of human hepatocyte nuclear factor 4 alpha (HNF4A) via AAV vectors or mRNA containing LNP. This type of gene therapy was able to decrease the expression of genes involved in profibrogenic activity and revert fibrosis in several mouse models with induced or genetic cholestasis [100]. 

#### 3.1.4. Amelioration of Bile Flow 

Finally, Adv-mediated hepatic delivery of aquaporin-1 (AQP1) has shown an improvement in the bile flow of estrogen-induced cholestatic rats [101]. In fact, this approach resulted in a marked reduction of serum ALP, as well as serum and biliary concentrations of bile salts. Moreover, AQP1 gene transfer increased biliary output as mediated by a significant increase in BSEP transport activity [102]. 

Thus, gene therapy approaches may offer a new avenue for the development of novel treatments for acquired cholestatic disorders.

### 3.2. Gene Therapy for Inherited Cholestasis

Gene therapy for the treatment of inherited hepatic diseases has garnered a great deal of attention after demonstrating that AAV vectors expressing human coagulation factors IX and VIII in the livers of patients with hemophilia B and A, respectively, resulted in a sustained therapeutic effect for more than three years [103]. In fact, a large number of gene therapy products have demonstrated promising therapeutic effects in clinically relevant animal models, leading to clinical trials for inherited liver disorders, such as phenylketonuria, familial hypercholesterolemia, ornithine transcarbamylase deficiency, acute intermittent porphyria, methylmalonic acidemia, and Wilson’s disease, among others [88]. In the next sections of the review, we will focus on the use of gene therapy for inherited cholestatic diseases, which include genetic disorders with associated cholestasis and the different forms of PFIC.

#### 3.2.1. Gene Therapy of Genetic Disorders with Associated Cholestasis

Preclinical studies have shown promising results in animal models of Cerebrotendinous xanthomatosis (CTX) and Crigler-Najjar syndrome type 1. In the first case, the administration of an AAV8 vector expressing CYP27A was able to restore BA metabolism and normalize the concentration of most BAs in plasma in a mouse model of CTX [104]. Interestingly, this therapeutic effect was achieved with only 20% of transduced hepatocytes, which could greatly facilitate the clinical translation of this approach. Secondly, treatment of Crigler–Najjar syndrome type 1 with an AAV8 vector expressing UDP-glucuronosyltransferase family 1-member A1 (UGT1A) showed normalization of total serum bilirubin levels in two animal models of the disease, Gunn rats and *Ugt1a1^-/-^* mice [105]. In this last model, a therapeutic effect was also demonstrated in newborn mice, although high doses of vector were required to maintain the effect [106]. These preclinical results led to a phase I/II clinical trial sponsored by Genethon (Évry, France), which is currently ongoing (NCT03466463).

The results observed in preclinical studies of Crigler–Najjar syndrome showed that one of the main limitations for gene therapy of genetic cholestatic diseases could be related to the loss of viral genomes associated with hepatocyte proliferation occurring in young patients [107]. 

#### 3.2.2. Gene Therapy for PFIC Diseases

Gene therapy approaches for PFIC can be based on gene supplementation or gene editing strategies to modify and repair the affected genes. The implementation of gene therapy for the different types of PFIC has some limitations. Firstly, in some types of PFIC in order to achieve stable and long-term therapeutic efficacy, it could be necessary to transduce most of the hepatocytes, which may require the use of high doses of the viral vector with the concomitant safety concerns [107,108]. Secondly, some types of PFIC have extrahepatic clinical manifestations hampering the liver-targeted treatment [109]. Finally, PFIC diseases requiring therapy are generally diagnosed in pediatric patients, and gene therapy based on non-integrative vectors, such as AAV, may be inefficient due to the loss of viral genomes associated with hepatocyte proliferation in a growing liver [107]. The decision to undergo gene therapy for PFIC, as well as the outcome of the therapy, will likely be influenced by the type of mutations present in the affected gene. For example, patients with missense mutations leading to decreased protein activity will probably respond better than those with a complete deficiency.

Although the loss of viral genomes could be a problem for most inherited cholestasis, ABCB4 deficiency, which causes PFIC3, has certain advantages over other PFIC types for liver gene therapy. For example, previous results using hepatocyte transplantation in a mouse model of PFIC3 showed that engraftment of 12% of healthy hepatocytes was enough to achieve therapeutic efficacy [110]. This evidence led to four preclinical studies examining the feasibility of gene therapy for PFIC3 in three different *Abcb4^-/-^* mouse models with a range of phenotypes depending on the mouse strain [111].

##### Gene Therapy for PFIC3 Based on ABCB4 Supplementation

The first study tested gene therapy in C57BL/6 *Abcb4^-/-^* mice that were challenged with a BA-enriched diet to increase liver toxicity due to their mild phenotype. Treatment with an AAV8 vector expressing ABCB4 demonstrated long-term efficacy by preventing the increase of serum transaminases and the loss of biliary PC levels after BA challenge [112]. In a second study, performed by our group, we evaluated PFIC3 AAV-based gene therapy in FVB *Abcb4^-/-^* mice, which have a clinically relevant phenotype characterized by high serum levels of bile salts and transaminases, hepatosplenomegaly, and liver fibrosis [113]. In this model, we demonstrated that an AAV8 vector containing a codon-optimized *ABCB4* sequence downstream of the liver specific alpha-1 antitrypsin (AAT) promoter resulted in stable and long-term correction of PFIC3 by improving all disease markers. Interestingly, this therapy was not only able to prevent disease progression in young mice (two-week-old), in which symptoms had not yet developed, but also in older mice with an established phenotype (five-week-old and sixteen-week-old mice). The therapeutic effect was dose dependent, and it was observed that restoration of biliary PC levels above 12–13% (over 4000 µM) of wild-type levels was enough to have a curative effect. This indicates that PFIC3 could be treated even if only a small fraction of hepatocytes were transduced, in this way resembling gene therapy of other diseases like hemophilia B, in which therapeutic effects can be obtained with a small percentage of transduced hepatocytes. In our study, the therapeutic threshold was achieved with as little as 2–3% of wild-type ABCB4 expression levels [113]. Interestingly, this therapy was more efficacious in male mice compared to females, although a sustained therapeutic effect could be obtained in females by the administration of a second vector dose [113]. 

Recently, a preclinical study based on LNP-encapsulated mRNA therapy was able to transiently reverse the disease phenotype in BALB/c *Abcb4^-/-^* mice [114]. BALB/c *Abcb4^-/-^* show similar levels of serum biomarkers as the FVB *Abcb4^-/-^* mice, but with a faster progression of liver fibrosis, leading to early development of primary liver cancers as well as an earlier onset of other complications, such as portal hypertension [111]. Five repeat *ABCB4* mRNA-LNP injections were able to restore ABCB4 expression and biliary PC levels (~42% of wild-type levels), as well as improve serum biomarker levels, liver fibrosis, and hepatomegaly [114,115]. However, these previously described non-integrative vector-based gene therapy strategies may have important limitations, such as loss of transgene expression, either because of loss of viral genomes due to hepatocyte division or because the short half-life of mRNA requires periodic administration to maintain the therapeutic effect. An alternative strategy to solve this hurdle is gene delivery mediated by an integrative vector. 

Using this type of approach, Siew et al. tested PFIC3 correction by the use of an integrative hybrid vector based on the expression of a piggyBac transposase and an AAV8 vector containing a piggyBac ABCB4 expression cassette in FVB *Abcb4^-/-^* mice. A single dose of the hybrid vector in neonates demonstrated the recovery of biliary PC levels and normalization of serum biomarkers. Additionally, the hybrid AAV-piggyBac treatment prevented biliary cirrhosis and reduced tumorigenesis [116]. However, the possibility of this vector integrating into oncogenic sites represents a high risk for clinical application. Results from these preclinical studies have led to orphan drug designation of an AAV vector harboring a codon optimized version of ABCB4 (VTX-803) developed by Vivet Therapeutics (Paris, France), opening a promising pathway for the treatment of patients with this cholestatic disorder (Table 2).

##### Gene Therapy for PFIC3 Targeting Mechanisms of Disease

Although gene supplementation or correction of the affected gene is the most straightforward gene therapy strategy for PFIC3, several studies have shown that it is also possible to treat this disease by altering the expression of other genes that are involved in this pathology. One example is the delivery of vectors that express genes that contribute to the attenuation of liver fibrosis, such as ACE2 and HNF4A, as described in Section 3.1.3. In this sense, an AAV8 vector expressing ACE2 was able to reduce liver fibrosis in early- and late-stage FVB *Abcb4^-/-^* mice [117]. Moreover, hepatocyte-targeted administration of *HNF4A* mRNA encapsulated with a biodegradable lipid restored the metabolic activity of hepatocytes in FVB *Abcb4^-/-^* mice, leading to a robust inhibition of fibrogenesis [100]. 

A novel approach that could be used to treat cholestatic diseases is based on the regulation of BA synthesis and homeostasis. It has recently been described that Limb expression 1-like protein (LIX1L) is increased in the liver of patients with cholestatic diseases and that the normalization of its expression alleviates cholestatic liver injury in different cholestatic mouse models, including FVB *Abcb4^-/-^* mice. LIX1L regulates the levels of miR-191-3p, a microRNA that downregulates transcription factor liver receptor homolog-1 (LRH-1), thereby inhibiting Cyp7a1 and Cyp8b1 expression, two enzymes required for BA synthesis. Based on these data, Li et al. [118], recently showed that an AAV vector overexpressing miR-191-3p was able to ameliorate cholestasis in FVB *Abcb4^-/-^* mice by direct repression of LRH-1 expression, thereby reducing de novo BA synthesis [118]. Another potential target for reducing liver fibrosis through gene therapy of cholestatic disorders is the suppression of the neurokinin 1 receptor (NK1R) axis as well as transforming growth factor-β1 (TGF-β1)/miR-31 signaling. In FVB *Abcb4^-/-^* mice, knock-out of NK1R has been shown to decrease the levels of miR-31 and of proinflammatory molecules such as TFG-β1, resulting in the reduction of liver inflammation and fibrosis [119]. These therapeutic approaches could be very useful for either acquired cholestatic disorders or PFIC.

##### Gene Therapy for Other Types of PFIC

For other types of PFIC, although gene supplementation using vectors expressing the specific mutated gene is also an option, there are certain barriers that make the development of these treatments more challenging than for PFIC3. For example, patients with PFIC1, PFIC4, PFIC5, and PFIC6 have extrahepatic manifestations that cannot be rescued by liver-targeted gene therapy [109,120]. In addition, in contrast to gene therapy for PFIC3, where liver toxicity arises in the bile canaliculi and transgene delivery to a fraction of hepatocytes leads to sufficient ABCB4 protein to reverse toxicity, in other types of PFIC where toxicity occurs in hepatocytes, it is likely that correction of a high percentage of these cells will be required to achieve a therapeutic effect [110,121]. One additional problem to develop gene therapy strategies for some types of PFIC is the lack of suitable animal models that adequately recapitulate the phenotype of patients. Currently, there are no *TJP2*-deficient animal models available to test the feasibility of gene therapy for PFIC4 [121]. Likewise, the existing animal model for PFIC6 is not suitable, because it has a complete knock-out of the MYO5B protein, which is not an appropriate model for this cholestatic disease. For that, it is necessary to develop an animal model with missense mutations of the *MYO5B* gene that affect the motor domain but do not result in complete deficiency of the protein [122]. In the case of PFIC2, there are several animal models that show a varying degree of pathology depending on the genetic background. *Abcb11^-/-^* mice in a C57BL/6 background represent the closest model to the patient disease phenotype, showing a drastic decrease in bile salt content in the bile that leads to increased levels of serum transaminases, liver fibrosis, and hepatomegaly, with these changes being more severe in females than in males [123]. However, unlike PFIC2 patients, these mice only show a mild elevation of serum bile salts, which is one of the main biomarkers of the disease. 

Finally, the loss of transgene expression by hepatocyte cell division is a drawback for the use of non-integrative vectors, such as AAV, in gene therapy of these inherited cholestatic disorders that need to be treated at very early ages, as only a few hepatocytes will maintain episomal AAV genomes [124]. Unlike PFIC3, for which partial gene therapy supplementation or correction of the affected gene is feasible, other types of PFIC may benefit from other gene therapy strategies aimed at reversing liver damage at several levels.

## 4. Future Directions

Due to the growing success of liver-targeted gene therapies and preclinical studies showing therapeutic efficacy against cholestatic diseases, such as PFIC3, the need to overcome challenges involved in taking these products from bench to bedside is even more critical.

One of the main challenges that gene therapy of cholestatic disorders faces is the potential loss of therapeutic effect in pediatric patients. This could be due to a decrease of viral genomes as a result of hepatocyte divisions in a growing liver in the case of AAV-based therapies, or to the transient expression of non-viral vector-mediated mRNA delivery [107,125]. Other challenges include immune responses to treatment (vector or transgene) and vector-mediated toxicities, particularly as a result of using very high vector doses. Strategies for addressing these challenges will guide the possible directions for present and future research. 

First, the administration of repeated doses of the vector could allow the maintenance of the therapeutic effect. However, this is only straightforward for non-viral vectors, such as mRNA-loaded nanoparticles, although it will greatly increase the cost of this therapy [115,125]. For viral vectors, such as AAV, the induction of vector-neutralizing antibodies after the first dose prevents the use of the same vector for additional administrations. However, several strategies have been proposed to allow vector re-administrations, which include the use of alternative AAV serotypes without cross-reactivity [126], the elimination of neutralizing antibodies using IgG-degrading endopeptidases [127], and the prevention of humoral and cellular responses against the virus via co-administration of the vector with rapamycin encapsulated in LNPs [128]. 

Second, the combination of gene therapy vectors with pharmacological therapies, such as UDCA, could provide synergistic therapeutic effects, especially in PFIC3 patients with more severe pathology who do not respond to UDCA treatment [19]. The use of pharmacological therapies in some pediatric patients could lead to a healthier liver status, improving the vector transduction efficiency and/or allowing the administration of the gene therapy product at an older age, at which vector genomes could be maintained for longer periods of time. 

Third, the sequential therapy of non-viral vectors such as mRNA-loaded nanoparticles in pediatric patients with growing livers followed by the administration of a viral vector that allows safe and stable long-term expression of the transgene at an older age, or the combination of vectors that, after reducing liver injury, facilitate the long-term efficacy of gene therapy could be of interest.

Fourth, the improvement of gene therapy vectors by codon optimization or incorporating promoters that allow a more potent expression of the transgene with the aim of reducing the viral dose required to achieve a therapeutic effect could function to reduce the risk of toxicity from high doses [129,130]. Alternatively, inducible promoters could allow a safe, precise, and controlled expression of the transgene with physiological transgene regulation [131,132], thus avoiding unwanted effects of transgene overexpression, such as silencing, exacerbated immune responses, or cytotoxicity that could result in the elimination of the transduced hepatocytes [133,134]. The modification of the transgene via codon optimization with a reduced number of CpG motifs may also mitigate the risk of activating the Toll-like receptor 9 pathway [135], which has been theorized to result in loss of transduced hepatocytes [136].

And finally, for those cholestatic disorders in which correction of the majority of hepatocytes for a long-term therapeutic effect is likely necessary, as in the case of some PFIC subtypes [108,109], a promising alternative is the use of CRISPR/Cas9 to achieve specific gene correction by non-homologous end-joining, base editing, or prime editing. The high efficiency of liver-targeted gene delivery makes it an ideal organ for the application of gene editing strategies in animal models of PFIC [88]. However, there are still many barriers hampering the use of gene editing techniques in humans, such as reduced specificity of targeted integration leading to safety concerns, as well as the low efficacy of non-homologous end-joining [137]. However, for most patients with more severe extrahepatic pathologies, liver-targeted gene therapy by itself will not be sufficient [109,122]. In this sense, the combination of gene therapy products targeting the liver with other therapies that allow the alleviation of extrahepatic damage could show a beneficial effect in these patients.

## 5. Conclusions

Although pharmacological therapies can be used to treat cholestatic diseases with milder phenotypes, they are less efficient in patients with a more severe pathology. As addressed in this review, alternative approaches, such as gene therapy, could represent a promising novel approach. To date, many preclinical studies using liver-directed gene therapy in clinically relevant animal models of both inherited and induced cholestasis have shown promising results. Although there are still many challenges for the implementation of these emerging treatments in the clinic, it is likely that some of these therapies will be approved in the near future, giving new hope for many cholestatic patients. 

## Figures and Tables

**Figure 1 biomedicines-10-01238-f001:**
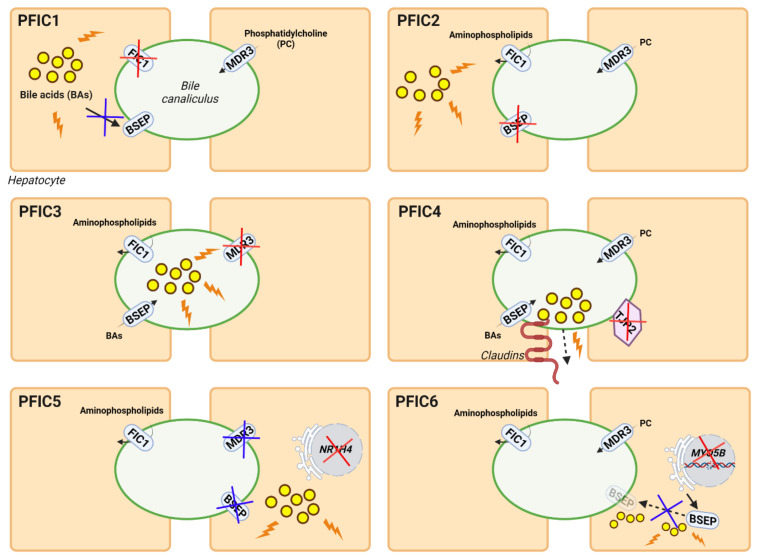
Genetic classification and pathogenesis of PFIC. The diagrams show the genes and functions altered in each type of PFIC. The main deficient proteins for each type of PFIC are indicated by red crosses, while derived alterations in other proteins or pathways are indicated by blue crosses. Damage due to the abnormal accumulation of BAs is shown as yellow circles with orange lightnings.

**Figure 2 biomedicines-10-01238-f002:**
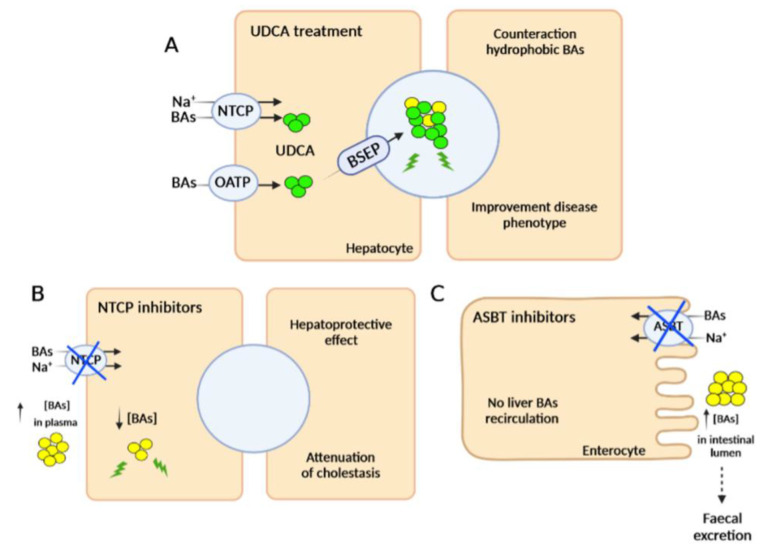
Pharmacological treatments for cholestatic diseases. (**A**) Mechanisms of action of UDCA, which favors the presence of hydrophilic BAs over hydrophobic BAs in bile, decreasing the toxic effect of “detergent bile” in cholestatic patients. (**B**) NTCP transporter inhibitors block the entry of BAs into hepatocytes. (**C**) ASBT inhibitors prevent the reabsorption of BAs in enterocytes, decreasing their entrance into the enterohepatic recirculation. Inhibitions are indicated with blue crosses. BA, bile acid (yellow circles).

**Figure 3 biomedicines-10-01238-f003:**
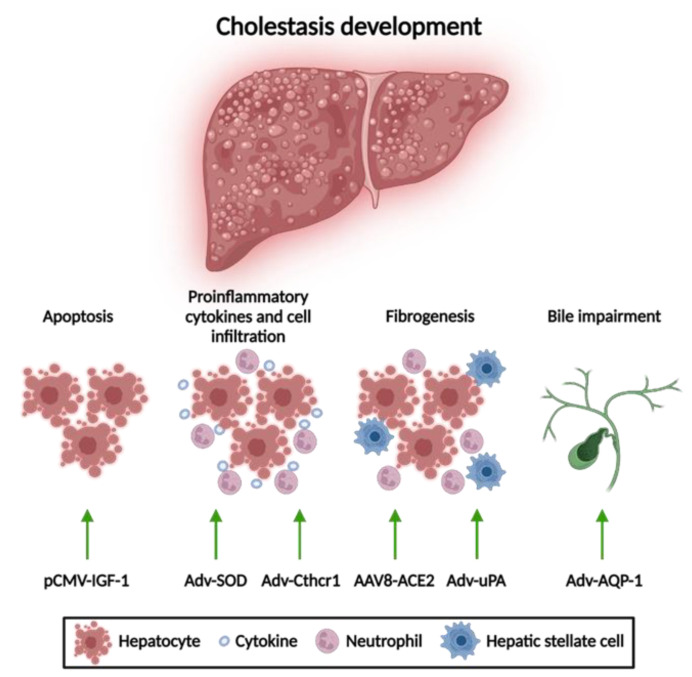
Gene therapy approaches for acquired cholestatic diseases. Different gene therapy strategies have resulted in an alleviation of liver disorders according to their anti-apoptotic, anti-inflammatory, and anti-fibrotic properties, respectively. Adv, adenoviral vector; AAV8, adeno-associated vector with serotype 8; ACE2, angiotensin-converting enzyme; AQP-1, aquaporin; Cthrc-1, collagen triple helix repeat containing-1; HNF4a, hepatocyte nuclear factor 4 alpha; IGF, insulin-like growth factor; SOD, superoxide dismutase; uPA, urokinase-plasminogen activator. This figure was created using BioRender.com.

**Table 1 biomedicines-10-01238-t001:** Drug therapy for cholestatic diseases in clinical trials.

		Drug Name	Indication	Current Status	Clinical Trial	Sponsor [Reference]
**FXR agonists**	Bile acids	UDCA (Actigall/Ursodiol/Ursofalk)	ICP	Phase III Phase IV	NCT01576458 NCT01510860	Turku University Hospital [68] Pharma GmbH [69]
			PBC	Approved		Sanofi-Synthelabo [70]
			PFIC3			[71]
		Nor-UDCA	PSC	Phase II	NCT01755507	Pharma GmbH [41]
		TUDCA (Taurolite)	PBC	Phase III	NCT01857284	Beijing Friendship Hospital [43]
		OCA (INT-747/Ocaliva)	PBC	Phase II Phase III	NCT00570765 NCT01473524	Intercept Pharmaceuticals [44,45,72,73]
			PSC	Phase II	NCT02177136	
	Non-bile acids	Cilofexor (CILO)	PSC	Phase I/II	NCT02943460	Gilead Sciences [49]
		Tropifexor (LJN452)	PBC	Phase II	NCT02516605	Novartis Pharmaceuticals [74]
		EDP-305	PBC	Phase II	NCT03394924	Enanta Pharmaceuticals
**ASBT inhibitors**		Odevixibat (A4250)	ALGS	Phase III	NCT04674761	Albireo [75,76]
			PFIC	Approved		
		Maralixibat (LUM001)	ALGS	Approved		Mirum Pharmaceuticals, Inc. [61]
			PFIC	Phase III	NCT02057718 NCT03905330	
		Linerixibat (GSK2330672)	PBC	Phase III	NCT02966834 NCT04167358	GlaxoSmithKline [77,78]
		Volixibat (SHP626)	ICP PBC PSC	Phase II	NCT04718961 NCT05050136 NCT04663308	Mirum Pharmaceuticals, Inc.
**Other pharmacotherapeutic agents**		Aldafermin (NGM282)	PBC	Phase II	NCT02026401	NGM Biopharmaceuticals, Inc. [79]
			PSC		NCT02704364	
		Bezafibrate	PBC	Phase III	NCT01654731	Hôpitaux de Paris [80]
		Elafibranor	PBC	Phase II	NCT03124108	Genfit [81]
		Seladelpar (MBX-8025)	PBC	Phase III	NCT03602560	CymaBay Therapeutics, Inc. [82]

**Table 2 biomedicines-10-01238-t002:**
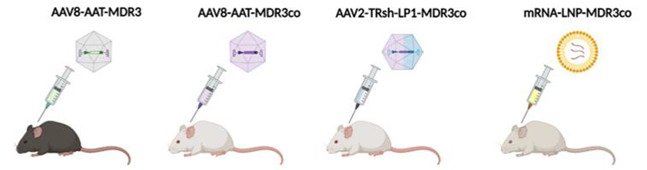
Gene therapy approaches for PFIC3.

	Aronson et al. [112]	Weber et al. [113]	Siew et al. [116]	Wei et al. [114]
**Strain Background**	C57BL/6 *Abcb4^-/-^*	FVB *Abcb4^-^*^/-^	FVB *Abcb4^-^*^/-^	BALB/c *Abcb4^-^*^/-^
**Phenotype**	Mild (requiring cholate-enriched diet)	Severe (similar to patients)	Severe (similar to patients)	More severe
**Vector**	AAV8	AAV8	Hybrid AAV-piggyBac transposon	LNP
**Dose**	5 × 10^13^ vg/kg	1 × 10^14^ vg/kg	~2 × 10^14^ vg/kg	1.0 mg/kg
**Age of treatment**	10-week-old	2- or 5-week-old	Newborn	4-week-old
**Outcomes**	Increased biliary PC and cholesterol content. Rescue of serum ALT, ALP and bilirubin levels. Prevention of liver fibrosis.	Increased biliary PC. Rescue of serum transaminases, ALP and BA levels. Improvement of the degree of hepatosplenomegaly. Prevention and reversal of liver fibrosis.	Increased biliary PC. Decreased hepatomegaly and serum parameters (ALT, ALP, BAs). Reduced liver fibrosis and liver tumor incidence.	Increased biliary PC (10–25% WT) and %BW. Decreased hepatomegaly and serum parameters (ALT, ALP, BAs). Normalization of liver fibrosis and portal hypertension.
**Advantages**	Long-term correction. No risk of mutagenesis.	Granted orphan drug designation. Long-term prevention and correction at early and late stages of PFIC3, respectively. No risk of mutagenesis.	Long-term correction. Preventing genome loss by hepatocellular proliferation during liver growth.	No risk of mutagenesis. Less immune responses.
**Disadvantages**	Need for challenge with BA-enriched dietary supplementation (model). Need to evaluate efficacy in younger mice more representative of the age of patients. Risks of using a high viral dose.	Loss of long-term therapeutic effect in half of the females treated with a single dose. Need to address the immune response based on anti-AAV neutralizing antibody for repeated administrations of the vector. Risks of using a high viral dose.	Risk of mutagenesis. Transposase overexpression Lack of serotype that efficiently transduces human hepatocytes.	Less durable expression. Requires repeated parenteral dosing.

AAT, alpha-1 antitrypsin; AAV, adeno-associated vector; ALP, alkaline phosphatase; ALT, alanine aminotransferase; BW, body weight; LNP, lipid nanoparticles; LP1, liver-specific transcriptional control unit; PC, phosphatidylcholine; TRsh, short piggyBac terminal repeats; VG, viral genomes; WT, wild-type.

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
