# Peer review of "Gene Therapy for Acquired and Genetic Cholestasis"

_biomedicines, 2022, doi:10.3390/biomedicines10061238_

Round 1

Reviewer 1 Report

This manuscript is a very clear and well organized review of the issues involved. The following points require correction.
Familial diseases are known to have different levels of protein expression depending on the type of gene mutation. For example, a nonsense mutation would have lost expression, as would a knockout mouse. On the other hand, in the case of a missense mutation, it is likely that a protein with reduced function is expressed. It seems necessary to explain the problem of gene therapy, especially when expression is not reduced.
Also, regarding side effects, it seemed to me that it would be good to describe the possibility that the expressed mRNA or the introduced RNA molecule could act as a sponge for untranslated RNA (e.g. microRNA) in the cell.

There was a place where it seemed like extra space was left out. Please correct.

Author Response

Dear Reviewer,

Please find below a point-by-point reply to each of your comments for the manuscript entitled “Gene Therapy for Acquired and Genetic Cholestasis”. All changes suggested by you have been highlighted in yellow in the manuscript.

Reviewer #1:

Familial diseases are known to have different levels of protein expression depending on the type of gene mutation. For example, a nonsense mutation would have lost expression, as would a knockout mouse. On the other hand, in the case of a missense mutation, it is likely that a protein with reduced function is expressed. It seems necessary to explain the problem of gene therapy, especially when expression is not reduced.

We thank this reviewer for bringing this point to our attention. We totally agree that the justification for as well as the outcome of gene therapy could be affected by the type of mutations present in the affected gene. In fact, this has been shown for pharmacological treatments based on UDCA, the first treatment option for PFIC3. Specifically, UDCA efficacy depends on the type of mutation, with patients having missense mutations showing a better response compared to those with complete ABCB4 deficiency. In order to clarify this point we have added the following sentence to the first paragraph of part 3.2.2: “The decision to undergo, as well as the outcome, of gene therapy for PFIC will likely be influenced by the type of mutations present in the affected gene. For example, patients with missense mutations leading to decreased protein activity will probably respond better than those having a complete deficiency.”

Also, regarding side effects, it seemed to me that it would be good to describe the possibility that the expressed mRNA or the introduced RNA molecule could act as a sponge for untranslated RNA (e.g. microRNA) in the cell.

It is true that overexpression of mRNAs in transduced cells could potentially act as a sponge for small untranslated RNAs, with possible deleterious effects. However, to our knowledge this effect has not been described in the context of gene supplementation therapy. One strategy by which this problem could be mitigated is through codon optimization, making the mRNA non-susceptible to binding by microRNAs present in target cells. However, since this problem has not been described so far in gene therapy approaches we believe that it is a secondary topic better discussed in more specialized articles that deal with gene regulation and/or the molecular complexities of gene therapy rather than here.

Reviewer 2 Report

Gene therapy is rapidly evolving therapeutic approach with high potential in medicine, especially in the treatment of rare diseases such as inherited cholestasis. In my opinion, topic of manuscript is suitable for the biomedicines. I appreciate, that authors discussed gene therapy of inherited cholestasis in the context of disease pathogenesis and other used/studied therapies. Nevertheless, for the improving quality of manuscript, I would like recommend incorporation of widely discussion and description of alone gene therapy (e.g., history, principle and used vectors, advanced and disadvantages, future development).  Besides, I have no serious objections against acceptation of this manuscript.

Author Response

Dear Reviewer,

Please find below a point-by-point reply to each of your comments for the manuscript entitled “Gene Therapy for Acquired and Genetic Cholestasis”. All changes suggested by you have been highlighted in yellow in the manuscript.

Reviewer #2:

For the improving quality of manuscript, I would like recommend incorporation of widely discussion and description of alone gene therapy (e.g., history, principle and used vectors, advanced and disadvantages, future development).  Besides, I have no serious objections against acceptation of this manuscript.

We have added a new paragraph at the beginning of Part 3 that contains a general description of gene therapy, as suggested by this reviewer. However, due to the focus of the review being on a more specific aspect within the field rather than a broader overview, we have kept this introduction brief. For readers more interested in basic concepts of gene therapy we have added a reference of a more general review about gene therapy.